

# Classification of movie reviews using term frequency-inverse document frequency and optimized machine learning algorithms

Muhammad Zaid Naeem[1,*], Furqan Rustam[1,*], Arif Mehmood[2], Mui-zzud-din [1], Imran Ashraf[3] and Gyu Sang Choi[3]

[1] Department of Computer Science, Khwaja Fareed University of Engineering and Information Technology, Rahim Yar Khan, Pakistan
[2] Department of Computer Science & Information Technology, The Islamia University of Bahawalpur, Bahawalpur, Pakistan
[3] Information and Communication Engineering, Yeungnam University, Gyeongsan si, Daegu, South Korea
* These authors contributed equally to this work.

Corresponding authors
Imran Ashraf,
imranashraf@ynu.ac.kr
Gyu Sang Choi, castchoi@ynu.ac.kr

## ABSTRACT

The Internet Movie Database (IMDb), being one of the popular online databases for movies and personalities, provides a wide range of movie reviews from millions of users. This provides a diverse and large dataset to analyze users' sentiments about various personalities and movies. Despite being helpful to provide the critique of movies, the reviews on IMDb cannot be read as a whole and requires automated tools to provide insights on the sentiments in such reviews. This study provides the implementation of various machine learning models to measure the polarity of the sentiments presented in user reviews on the IMDb website. For this purpose, the reviews are first preprocessed to remove redundant information and noise, and then various classification models like support vector machines (SVM), Naïve Bayes classifier, random forest, and gradient boosting classifiers are used to predict the sentiment of these reviews. The objective is to find the optimal process and approach to attain the highest accuracy with the best generalization. Various feature engineering approaches such as term frequency-inverse document frequency (TF-IDF), bag of words, global vectors for word representations, and Word2Vec are applied along with the hyperparameter tuning of the classification models to enhance the classification accuracy. Experimental results indicate that the SVM obtains the highest accuracy when used with TF-IDF features and achieves an accuracy of 89.55%. The sentiment classification accuracy of the models is affected due to the contradictions in the user sentiments in the reviews and assigned labels. For tackling this issue, TextBlob is used to assign a sentiment to the dataset containing reviews before it can be used for training. Experimental results on TextBlob assigned sentiments indicate that an accuracy of 92% can be obtained using the proposed model.

## INTRODUCTION

Social media has become an integral part of human lives in recent times. People want to share their opinions, ideas, comments, and daily life events on social media. In modern times, social media is used for showcasing one's esteem and prestige by posting photos, text, video clips, *etc.* The rise and wide usage of social media platforms and microblogging websites provide the opportunity to share as you like where people share their opinions on trending topics, politics, movie reviews, *etc.* Shared opinions on social networking sites are generally known as short texts (ST) concerning the length of the posted text (*Sahu & Ahuja, 2016*). ST has gained significant importance over traditional blogging because of its simplicity and effectiveness to influence the crowd. These ST take the form of jargon and are even used by search engines as queries. Apart from being inspiring, the ST contains users' sentiments about a specific personality, topic, or movie and can be leveraged to identify the popularity of the discussed item. The process of mining the sentiment from the texts is called sentiment analysis (SA) and has been regarded as a significant research area during the last few years (*Hearst, 2003*). Sentiments given on social media platforms like Twitter, Facebook, *etc.* can be used to analyze the perception of people about a personality, service, or product, as well as, used to predict the outcome of various social and political campaigns. Thus, SA helps to increase the popularity and followers of political leaders, as well as, other important personalities. Many large companies like Amazon, Apple, and Google use the reviews of their employees to analyze the response to various services and policies. In the business sector, companies use SA to derive new strategies based on customer feedback and reviews (*Hand & Adams, 2014*; *Alpaydin, 2020*).

Besides the social media platforms, several websites serve as a common platform for discussions about social events, sports, and movies, *etc.*, and the Internet Movie Database (IMDb) is one of the websites that offer a common interface to discuss movies and provide reviews. Reviews are short texts that generally express an opinion about movies or products. These reviews play a vital role in the success of movies or sales of the products (*Agarwal & Mittal, 2016*). People generally look into blogs, review sites like IMDb to know the movie cast, crew, reviews, and ratings of other people. Hence it is not only the word of mouth that brings the audience to the theaters, reviews also play a prominent role in the promotion of the movies. SA on movie reviews thus helps to perform opinion summarization by extracting and analyzing the sentiments expressed by the reviewers (*Ikonomakis, Kotsiantis & Tampakas, 2005*). Being said that the reviews contain valuable and very useful content, the new user can't read all the reviews and perceive the positive or negative sentiment. The use of machine learning approaches proves to ease this difficult task by automatically classifying the sentiments of these reviews. Sentiment classification involves three types of approaches including the supervised machine learning approach, using the semantic orientation of the text, and use of SentiWordNet based libraries (*Singh et al., 2013a*).

Despite being several approaches presented, several challenges remain unresolved to achieve the best possible accuracy for sentiment analysis. For example, a standard sequence

for preprocessing steps is not defined and several variations are used which tend to show slightly different accuracy. Bag of words (BoW) is widely used for sentiment analysis, however, BoW loses word order information. Investigating the influence of other feature extraction approaches is of significant importance. Deep learning approaches tend to show better results than the traditional machine learning classifiers, but the extent of their better performance is not defined. This study uses various machine learning classifiers to perform sentiment analysis on the movie reviews and makes the following contributions

- This study proposes a methodology to perform the sentiment analysis on the movie reviews taken from the IMDb website. The proposed methodology involves preprocessing steps and various machine learning classifiers along with several feature extraction approaches.
- Both simple and ensemble classifiers are tested with the methodology including decision trees (DT), random forest (RF), gradient boosting classifier (GBC), and support vector machines (SVM). In addition, a deep learning model is used to evaluate its performance in comparison to traditional machine learning classifiers.
- Four feature extraction techniques are tested for their efficacy in sentiment classification. Feature extraction approaches include term frequency-inverse document frequency (TF-IDF), BoW, global vectors (GloVe) for word representations, and Word2Vec.
- Owing to the influence of the contradictions in users' sentiments in the reviews and assigned labels on the sentiment classification accuracy, in addition to the standard dataset, TextBlob annotated dataset is also used for experiments.
- The performance of the selected classifiers is analyzed using accuracy, precision, recall, and F1 score. Additionally, the results are compared with several state-of-the-art approaches to sentiment analysis.

The rest of this paper is organized as follows. "Related Work" discusses a few research works which are closely related to the current study. The selected dataset, machine learning classifiers, and preprocessing procedure, and the proposed methodology are described in "Materials and Methods". Results are discussed in "Results and Discussion" and finally, "Conclusion" concludes the paper with possible directions for future research.

## RELATED WORK

A large amount of generated data on social media platforms on Facebook, and Twitter, *etc.* are generating new opportunities and challenges for the researchers to *fetc*h useful and meaningful information to thrive business communities and serve the public. As a result, multidimensional research efforts have been performed for sentiment classification and analysis. Various machine learning and deep learning approaches have been presented in the literature in this regard. Few research works which is related to the current study are discussed here; we divide the research works into two categories: machine learning approaches and deep learning approaches.

The use of machine learning algorithms has been accelerated in several domains including image processing, object detection and natural language processing tasks, *etc.* (*Ashraf et al., 2019a*; *Khalid et al., 2020*; *Ashraf et al., 2019b*). For example, The study (*Hakak et al., 2021*) uses a machine learning approach for the fake new classification. The study proposes a feature selection technique and an ensemble classifier using three machine learning classifiers including DT, RF, and extra tree classifier. The proposed model achieves a good accuracy score on the 'Liar dataset' as compared to the ISOT dataset.

The authors implement several machine learning classification models for sentiment classification of IMDb reviews into positive and negative sentiments in *Pang, Lee & Vaithyanathan (2002)*. For this purpose, a dataset containing 752 negative reviews and 1,301 positive reviews from the IMDb website is used. The research aims at finding the suitable model with the highest F1 score and best generalization. Various combinations of features and hyperparameters are used for training the classifiers for better accuracy. K-fold cross-validation is used for evaluating the performance of the classifiers. Naive Bayes tend to achieve higher accuracy of 89.2% than the SVM classifier which achieves 81.0% accuracy.

Similarly, the study (*Singh et al., 2013a*) conducts experimental work on performance evaluation of the SentiWordNet approach for classification of movie reviews. The SentiWordNet approach is implemented with different variations of linguistic features, scoring schemes, and aggregation thresholds. For evaluation, two large datasets of movie reviews are used that contain the posts on movies about revolutionary changes in Libya and Tunisia. The performance of the SentiWordNet approach is compared with two machine learning approaches including NB and SVM for sentiment classification. The comparative performance of the SentiWordNet and machine learning classifiers show that both NB and SVM perform better than all the variations of SentiWordNet.

A hybrid method is proposed in *Singh et al. (2013b)* where the features are extracted by using both statistical and lexicon methods. In addition, various feature selection methods are applied such as Chi-Square, correlation, information gain, and regularized locality preserving indexing (RLPI) for the features extraction. It helps to map the higher dimension input space to the lower dimension input space. Features from both methods are combined to make a new feature set with lower dimension input space. SVM, NB, K-nearest neighbor (KNN), and maximum entropy (ME) classifiers are trained using the IMDb movie review dataset. Results indicate that using hybrid features of TF and TF-IDF with Lexicon features gives better results.

The authors propose an ensemble approach to improve the accuracy of sentiment analysis in *Minaee, Azimi & Abdolrashidi (2019)*. The ensemble model comprises convolutional neural network (CNN) and bidirectional long short term memory (Bi-LSTM) networks and the experiments are performed on IMDb review and Stanford sentiment treebank v2 (SST2) datasets. The ensemble is formed using the predicted scores of the two models to make the final classification of the sentiment of the reviews. Results indicate that the ensemble approach performs better than the state-of-the-art approaches and achieves an accuracy of 90% to classify the sentiment from reviews.

The authors investigate the use of three deep learning classifiers including multilayer perceptron, CNN, and LSTM for sentiment analysis in *Ali, Abd El Hamid & Youssif (2019)*. Besides, experiments are also carried using a hybrid model CNN-LSTM for sentiment classification, and the performance of these models is compared with support vector machines and Naive Bayes. Multilayer Perceptron (MLP) is developed as a baseline for other networks' results. LSTM network, CNN, and CNN-LSTM are applied on the IMDb dataset consisting of 50,000 movies reviews. The word2vec is applied for word embedding. Results indicate that higher accuracy of 89.2% can be achieved from the hybrid model CNN-LSTM. Individual classifiers show a lower accuracy of 86.74%, 87.70%, and 86.64% for MLP, CNN, and LSTM, respectively.

Similarly, an ensemble classifier is proposed in *Yenter & Verma (2017)* which comprises CNN and LSTM networks. The model aims at the word-level classification of the IMDb reviews. The output of the CNN network is fed into an LSTM before being concatenated and sent to a fully connected layer to produce a single final output. Various regularization techniques, network structures, and kernel sizes are used to generate five different models for classification. The designed models can predict the sentiment polarity of IMDb reviews with 89% or higher accuracy.

The study (*Jain & Jain, 2021a*) conducts experiments using the IMDb review dataset with deep learning models for sentiment classification. It uses a convolutional neural network (CNN) and long short-term memory (LSTM) with different activation functions. The highest accuracy of 0.883 is achieved with CNN using the ReLU activation function. Similarly (*Nafis & Awang, 2021*) proposes a hybrid approach for IMDb review classification using TF-IDF and SVM. The approach called SVM-RFE uses important feature selection to train the SVM model. Feature selection helps in boosting the performance of SVM and increases the accuracy to 89.56% for IMDb reviews sentiment classification. The study (*Giatsoglou et al., 2017*) proposed an approach for sentiment analysis using a machine learning model. A hybrid feature vector is proposed by combining word2vec and BoW technique and experiments are performed using four datasets containing online user reviews in Greek and English language. In a similar fashion (*Mathapati et al., 2018*) performs sentiment analysis on IMDb reviews using a deep learning approach. The study used a CNN and LSTM recurrent neural network to obtain significant accuracy on the IMDb reviews dataset.

The study (*Jain & Jain, 2021b*) uses a machine learning approach for IMDb reviews classification. The study performs preprocessing of data and proposes a feature selection technique using association rule mining (ARM). Results show that Naive Bayes (NB) outperforms all other used models by achieving a 0.784 accuracy score using the proposed features. The study (*Qaisar, 2020*) presents an approach using LSTM for IMDb review sentiment classification. LSTM achieves an 0.899 accuracy score on the IMDb dataset. Along the same lines (*Shaukat et al., 2020*) performs experiments on the IMDb reviews dataset using a supervised machine learning approach. The study proposed neural network can achieve a 0.91 accuracy score.

From the above-discussed research works, it can be inferred that supervised machine and deep learning approaches show higher performance than lexicon-based approaches.

**Table 1 Comprehensive summary of research works discussed in the related work.**

| Reference | Approach | Model | Aim |
|---|---|---|---|
| Singh et al. (2013a) | Lexicon-Based | SentiWordNet | Movie review classification |
| Singh et al. (2013b) | Machine Learning | RLPI, Hybrid Features, KNN | IMDb reviews classification |
| Yenter & Verma (2017) | Deep Learning | CNN LSTM | IMDb reviews classification |
| Giatsoglou et al. (2017) | Machine Learning | BoW-DOUBLE and Average emotion-DOUBLE | IMDb reviews classification |
| Mathapati et al. (2018) | Deep Learning | CNN | IMDb reviews classification |
| Ali, Abd El Hamid & Youssif, 2019 | Deep Learning | Multilayer perceptron, CNN and LSTM | IMDb reviews classification |
| Minaee, Azimi & Abdolrashidi, 2019 | Deep Learning | Bi-LSTM | IMDb review and Stanford sentiment treebank v2 (SST2) |
| Qaisar (2020) | Deep Learning | LSTM | IMDb reviews classification |
| Shaukat et al. (2020) | Deep & Machine Learning | NN | IMDb reviews classification |
| Jain & Jain (2021a) | Deep Learning | CNN | IMDb reviews classification |
| Nafis & Awang (2021) | Machine Learning | SVM + (SVM-RFE) | IMDb reviews classification |
| Jain & Jain (2021b) | Machine Learning | NB + ARM | IMDb reviews classification |

**Table 2 Description of IMDb dataset variables.**

| Review | Label |
|---|---|
| Gwyneth Paltrow is absolutely great in this mo… | 0 |
| I own this movie. Not by choice, I do. I was r… | 0 |
| Well I guess it supposedly not a classic becau… | 1 |
| I am, as many are, a fan of Tony Scott films… | 0 |
| I wish "that '70s show" would come back on tel… | 1 |

Additionally, the accuracy offered by machine learning approaches requires further improvement, as shown in Table 1. This study focus on using several machine learning classifiers for this purpose, in addition to three feature extraction, approaches for enhanced classification performance. This study contributes to filling the literature gap which is accuracy and efficiency for IMDb review sentiment classification using state-of-the-art techniques.

# MATERIALS AND METHODS

This section describes the dataset used for the experiments, machine learning classifiers selected for review classification, as well as, the proposed methodology and its working principles.

## Data description

This study uses the 'IMDb Reviews' from Kaggle which contains users' reviews about movies (DAT, 2018). The dataset has been largely used for text mining and consists of reviews of 50,000 movie reviews of which approximately 25,000 reviews belong to the positive and negative classes, respectively. Table 2 shows samples of reviews from both negative and positive classes.

| Table 3 Contradiction in TextBlob and original dataset labels. | | |
|---|---|---|
| **Review** | **TextBlob** | **Original** |
| Movie makers always author work mean yes things condensed sake viewer interest look Anne Green gables wonderful job combining important events cohesive whole simply delightful believe chose combine three novels together Anne Avonlea dreadful mess look missed Paul Irving little Elizabeth widows windy poplars Anne college years heaven sake delightful meet Priscilla rest redmond gang Kevin Sullivan taken things one movie time instead jumbling together combining characters events way movie good leave novels montgomery beautiful work something denied movie let seeing successful way brough Anne green gables life | Positive | Negative |

## TextBlob

TextBlob is a Python library that we used to annotate the dataset with new sentiments (*Tex, 2020*; *Loria, 2018*). TextBlob is used for labeling as the possibility of contradiction between the review text and label can not be ignored. TextBlob finds the polarity score for each word and then sums up these polarity scores to find the sentiment. TextBlob assigns a polarity score between −1 and 1. A polarity score greater than 0 shows the positive sentiment, a polarity score less than 0 shows a negative sentiment while a 0 score indicates that the sentiment is neutral. In the dataset used in this study, 23 neutral sentiments are found after applying TextBlob. Pertaining to the low number of neutral sentiments which can cause class imbalance, only negative and positive sentiments are used for experiments. Contradiction in TextBlob annotated label and original dataset label is shown in Table 3.

## Feature engineering methods

Identification of useful features from the data is an important step for the better training of machine learning classifiers. The formation of secondary features from the original features enhances the efficiency of machine learning algorithms (*Oghina et al., 2012*; *Mujahid et al., 2021*). It is one of the critical factors to increase the accuracy of the learning algorithm and boost its performance. The desired accuracy can be achieved by excluding the meaningless and redundant data. Less quantity of meaningful data is better than having a large quantity of meaningless data (*Prabowo & Thelwall, 2009*). So, feature engineering is the process of extracting meaningful features from raw data which helps in the learning process of algorithms and increases its efficiency and consistency (*Lee, Cui & Kim, 2016*).

### Bag of words

The BoW is simple to use and easy to implement for finding the features from raw text data (*Rustam et al., 2021*; *Rupapara et al., 2021a*). Many language modeling and text classification problems can be solved using the BoW features. In Python, the BoW is implemented using the CountVectorizer. BoW counts the occurrences of a word in the given text and formulates a feature vector of the whole text comprising of the counts of each unique word in the text. Each unique word is called 'token' and the feature vector is the matrix of these tokens (*Liu et al., 2008*). Despite being simple, BoW often surpasses many complicated feature engineering approaches in performance.

## Term frequency-inverse document frequency

TF-IDF is another feature engineering method that is used to extract features from raw data. It is mostly deployed in areas like text analysis and music information retrieval (*Yu, 2008*). In this approach, weights are assigned to every term in a document based on term frequency and inverse document frequency (*Neethu & Rajasree, 2013*; *Biau & Scornet, 2016*). Terms having higher weights are supposed to be more important than terms having lower weights. The weight for each term is based on the Eq. (1).

$$W_{i,j} = TF_{t,d}\left(\frac{N}{D_t}\right) \tag{1}$$

where $TF_{t,d}$ is the number of occurrences of term $t$ in document $d$, $D_{f,t}$ is the number of documents having the term $t$ and $N$ is the total number of documents in the dataset.

TF-IDF is a kind of scoring measurement approach which is widely used in summarization and information retrieval. TF calculates the frequency of a token and gives higher importance to more common tokens in a given document (*Vishwanathan & Murty, 2002*). On the other hand, IDF calculates the tokens which are rare in a *corpus*. In this way, if uncommon words appear in more than one document, they are considered meaningful and important. In a set of documents $D$, IDF weighs a token $x$ using the Eq. (2).

$$IDF(x) = N/n(x) \tag{2}$$

where $n(x)$ denotes frequency of $x$ in $D$ and $N/n(x)$ denotes the inverse frequency. TF-IDF is calculated using TF and IDF as shown in Eq. (3).

$$TF - IDF = TF \times IDF \tag{3}$$

TF-IDF is applied to calculate the weights of important terms and the final output of TF-IDF is in the form of a weight matrix. Values gradually increase to the count in TF-IDF but are balanced with the frequency of the word in dataset (*Zhang et al., 2008*).

## Word2Vec

Word2Vec is one of the widely used NLP techniques for feature extraction in text mining that transforms text words into vectors (*Wang, Ma & Zhang, 2016*). Given a *corpus* of text, Word2Vec uses a neural network model for learning word associations. Each unique word has an associated list of numbers called 'vector'. The cosine similarity of the vectors represents the semantic similarity between the words that are represented by vectors.

## GloVe

GloVe from Global Vectors is an unsupervised model used to obtain words' vector representation (*Bhoir, Ghorpade & Mane, 2017*). The vector representation is obtained by mapping the words in a space such that the distance between the words represents the semantic similarity. Developed at Stanford, GloVe can determine the similarity between words and arrange them in the vectors. The output matrix by the GloVe gives vector space of word with linear substructure.

**Table 4 Hyperparameters used for optimizing the performance of models.**

| Model | Hyperparameters | Values range used for tuning |
|---|---|---|
| RF | n_estimators = 300, random_state = 50, max_depth = 300 | n_estimators = {50 to 500}, random_state = {2 to 60}, max_depth = {50 to 500} |
| SVM | kernel= 'linear', C = 3.0, random_state = 50 | Kernel = {'linear' 'poly', 'sigmoid'}, C = {1.0 to 5.0}, random_state = {2 to 60} |
| DT | random_state = 50, max_depth = 300 | random_state = {2 to 60}, max_depth = {50 to 500} |
| GBC | n_estimators = 300, random_state = 50, max_depth = 300, learning_rate = 0.2 | n_estimators = {50 to 500}, random_state = {2 to 60}, max_depth = {50 to 500}, learning_rate = {0.1 to 0.8} |

## Supervised machine learning models

Several machine learning classifiers have been selected for evaluating the classification performs in this study. A brief description of each of these classifiers is provided in the following sections.

### Random forest

Rf is based on combining multiple decision trees on various subsamples of the dataset to improve classification accuracy. These subsamples are the combination of randomly selected features which are the size of the original dataset to form a bootstrap dataset. The average of predictions from these models is used to obtain a model with low variance. Information gain ratio and Gini index are the most frequently used feature selection parameters to measure the impurity of feature (*Agarwal et al., 2011*).

$$\sum \sum_{j \neq i} \left( \frac{f(C_i, T)}{|T|} \right) \left( \frac{f(C_j, T)}{|T|} \right) \tag{4}$$

where $\frac{f(C_i, T)}{|T|}$ indicates the probability of being a member of class $C_i$.

The decision trees are not pruned upon traversing each new training data set. The user can define the number of features and number of trees on each node and set the values of other hyperparameters to increase the classification accuracy (*Biau & Scornet, 2016*).

### Gradient boosting classifier

GBC is an ensemble classifier used for classification tasks with enhanced accuracy based on boosting (*Ayyadevara, 2018*). It combines many weak learners sequentially to reduce the error gradually. This study uses the GBC with decision tree as a weak learner. GBC performance depends on the loss function and mostly the logarithmic loss function is used for classification. In addition, weak learners and adaptive components are important parameters of GBC. The hyperparameters setting of GBC used in this study is shown in Table 4. GBC is used with 300 n_estimators indicating that 300 weak learners (decision trees) are combined under boosting method and each tree is restricted to 300 max_depth. The learning_rate is set to 0.2 which helps to reduce model overfitting (*Rustam et al., 2020*).

### Decision tree

DT is one of the most commonly used models for classification and prediction problems. DT is a simple and powerful tool to understand data features and infer decisions. The

decision trees are constructed by repeatedly dividing the data according to split criteria. There are three types of nodes in a decision tree: root, internal, and leaf. The root node has no incoming but zero or more outgoing edges, the internal node has exactly one incoming but two or more outgoing edges while the leaf node has one incoming while no outgoing edge (*Bakshi et al., 2016*; *Tan, Steinbach & Kumar, 2006*). Nodes and edges represent features and decisions of a decision tree, respectively. A decision tree can be binary or non-binary depending upon the leaves of a node. The gain ratio is one of the commonly used split criteria for DT.

$$Gain\ ratio = \frac{\Delta_{info}}{Split\ Info} \tag{5}$$

where split info is defined as in Eq. (6).

$$Split\ Info = -\sum_{i=1}^{k} P(v_i)log_2P(v_i) \tag{6}$$

where $k$ indicates the total number of splits for DT which is hyperparameter tuned for different datasets to elevate the performance. DT is non-parametric, computationally inexpensive, and shows better performance even when the data have redundant attributes.

### Support vector machine

Originally proposed by *Cortes & Vapnik (1995)* for binary classification, SVM is expanded for multi-class classification. SVM is a widely used approach for non-linear classification, regression, and outlier detection (*Bennett & Campbell, 2000*). SVM has the additional advantage of examining the relationship theoretically and performs distinctive classification than many complex approaches like neural networks (*Agarwal & Mittal, 2016*). SVM separates the classes by distinguishing the optimal isolating lines called hyperplane by maximizing the distance between the classes' nearest points (*Neethu & Rajasree, 2013*). Different kernels can be used with SVM to accomplish better performance such as radial, polynomial, neural, and linear (*Guzman & Maalej, 2014*). SVM is preferred for several reasons including the lack of local minimal, structural risk minimization principle, and developing more common classification ability (*Visa et al., 2011*; *Vishwanathan & Murty, 2002*).

For optimizing the performance of the machine learning models used in this study, several hyperparameters have been fine-tuned according to experience from the literature on text classification tasks. A list of the parameters and corresponding values used for experiments in this study is provided in Table 4.

## Proposed methodology

With the growing production of movies over the last two decades, a large number of opinions and reviews are posted on various social media platforms and websites. Such reviews are texts that show explicit opinions about a film or product. These opinions play an important part in the success of film or sales of the products (*Agarwal & Mittal, 2016*). People search blogs, and evaluation sites like IMDb to get the likes and dislikes of other

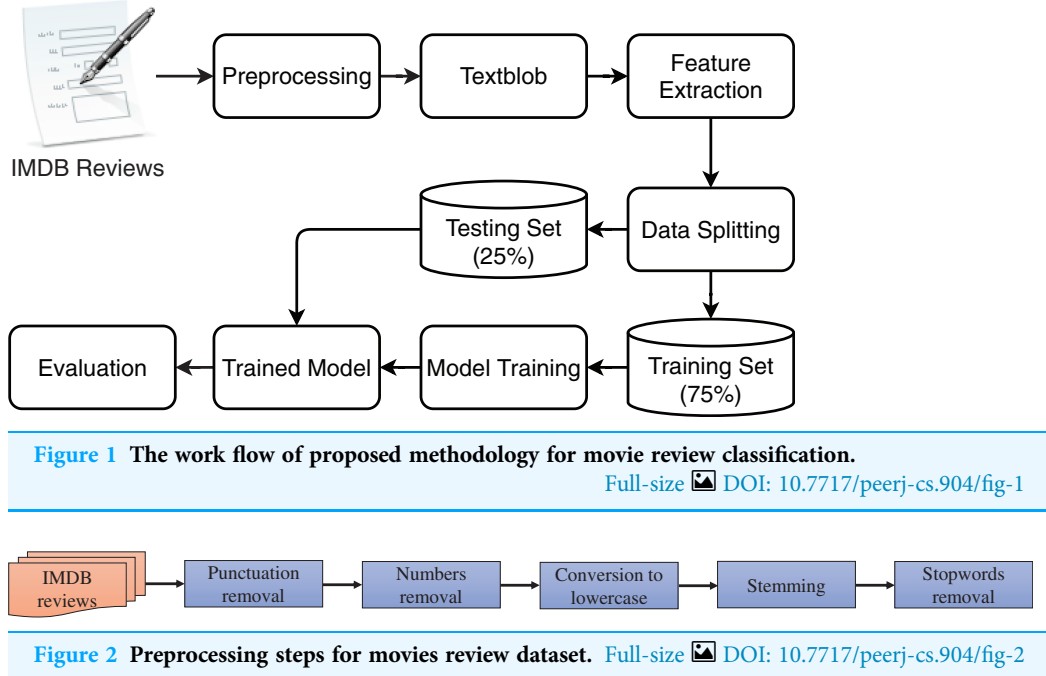

**Figure 1** The work flow of proposed methodology for movie review classification.

**Figure 2** Preprocessing steps for movies review dataset.

people about films, the cast, and team, *etc*. but it is very difficult to read every review and comment. Evaluation of these sentiments becomes beneficial to assisting people in this task. Sentiments expressed in such reviews are important regarding the evaluation of the movies and their crew. Automatic sentiment analysis with higher accuracy is extremely important in this regard and this study follows the same direction and proposes an approach to perform the sentiment analysis of movie reviews. In addition, since the contradictions in the expressed sentiments in movie reviews and their assigned labels can not be ignored, this study additionally uses TextBlob to determine the sentiments. Two sets of experiments are performed using the standard dataset and TextBlob annotated dataset to fill in the research gap as previous studies do not consider the contradictions in the sentiments and assigned labels. Figure 1 shows the flow of the steps carried out for sentiment classification.

As a first step, the reviews are preprocessed using a sequence of operations. Preprocessing is critical to boosting the training of the classifiers and enhancing their performance. The purpose of preprocessing is to clean the data by removing unnecessary, meaningless, and redundant text from reviews. For this purpose, the six steps are carried sequentially, as shown in Fig. 2.

Punctuation is removed from IMDb text reviews because punctuation does not add any value to text analysis (*Guzman & Maalej, 2014*). Sentences are more readable for humans due to punctuation, however, it is difficult for a machine to distinguish punctuation from other characters. Punctuation distorts a model's efficiency to distinguish between entropy, punctuation, and other characters (*Rupapara et al., 2021b*, *Liu et al., 2008*). Punctuation is removed from the text in pre-processing to reduce the complexity of

**Table 5  Text from sample review before and after punctuation removal.**

| Before punctuation removal | After punctuation removal |
|---|---|
| @Gwyneth Paltrow is absolutely… !!!great in this movie | Gwyneth Paltrow is absolutely great in this movie |
| I own this movie. This is number 1 movie… I didn't like by choice, I do | I own this movie This is number 1 movie I didnt like by choice I do |
| I wish "that '70s show" would come back on tel | I wish that 70s show would come back on tel |

**Table 6  Sample text from movie reviews after removing numeric values.**

| Input data | After numeric removal |
|---|---|
| Gwyneth Paltrow is absolutely great in this movie. | Gwyneth Paltrow is absolutely great in this movie |
| I own this movie This is number 1 movie I didnt like by choice I do. | I own this movie This is number movie I didnt like by choice I do |
| I wish that 70s show would come back on tel. | I wish that s show would come back on tel |

**Table 7  Sample output of the review text after changing the case of review text.**

| Input data | After case lowering |
|---|---|
| Gwyneth Paltrow is absolutely great in this movie. | gwyneth paltrow is absolutely great in this movie |
| I own this movie This is number movie I didnt like by choice I do. | i own this movie this is number movie i didnt like by choice i do |
| I wish that s show would come back on tel. | i wish that s show would come back on tel |

the feature space. Table 5 shows the text of a sample review, before and after the punctuation has been removed.

Once the punctuation is removed, the next step is to find numerical values and remove them as they are not valuable for text analysis. Numerical values are used in the reviews as an alternative to various English words to reduce the length of reviews and ease of writing the review. For example, 2 is used for 'to' and numerical values are used instead of counting like 1 instead of 'one'. Such numerals are convenient for humans to interpret, yet offer no help in the training of machine learning classifiers. Table 6 shows text from sample reviews after the numeric values are removed.

In the subsequent step of numbers removal, all capital letters are converted to lower form. Machine learning classifiers can not distinguish between lower and upper case letters and consider them as different letters. For example, 'Health', and 'health' are considered as two separate words if conversion is not performed from uppercase to lowercase. This may reduce the significance of most occurred terms and degrade the performance (*Liu & et, 2010*). It increases the complexity of the feature space and reduces the performance of classifiers; therefore, converting the upper case letters to lower form helps in increasing the training efficiency of the classifiers. Table 7 shows the text after the case is changed for the reviews.

Stemming is an important step in pre-processing because eliminating affixes from words and changing them into their root form is very helpful to enhance the efficiency of a model (*Goel, Gautam & Kumar, 2016*). For example, 'help', 'helped', and 'helping' are

**Table 8  Text from sample review before and after stemming.**

| Input data | After stemming |
|---|---|
| gwyneth Paltrow is absolutely great in this movie. | gwyneth paltrow is absolute great in this movie |
| i own this movie this is number movie i didnt like by choice I do. | i own this movie this is number movie i didnt like by choice i do |
| i wish that s show would come back on tel. | i wish that s show would come back on tel |

**Table 9  Sample reviews before and after the stop words removal.**

| Input data | After stopwords removal |
|---|---|
| gwyneth Paltrow is absolutely great in this movie. | gwyneth paltrow absolute great movie |
| i own this movie this is number movie i didnt like by choice I do. | own movie number movie didnt like choice do |
| i wish that s show would come back on tel. | wish show would come back tel |

**Table 10  BoW features from the preprocessed text of sample reviews.**

| No. | absolute | back | choice | come | didnt | do | great | gwyneth | like |
|---|---|---|---|---|---|---|---|---|---|
| 1 | 1 | 0 | 0 | 0 | 0 | 0 | 1 | 1 | 0 |
| 2 | 0 | 0 | 1 | 0 | 1 | 1 | 0 | 0 | 1 |
| 3 | 0 | 1 | 0 | 1 | 0 | 0 | 0 | 0 | 0 |
| **No.** | **movie** | **number** | **own** | **paltrow** | **show** | **tel** | **wish** | **would** | |
| 1 | 1 | 0 | 0 | 1 | 0 | 0 | 0 | 0 | |
| 2 | 2 | 1 | 1 | 0 | 0 | 0 | 0 | 0 | |
| 3 | 0 | 0 | 0 | 0 | 1 | 1 | 1 | 1 | |

altered forms of 'help', however, machine learning classifiers consider them as two different words (*Singh et al., 2013b*). Stemming changes these different forms of words into their root form. Stemming is implemented using the PorterStemmer library of Python (*Pang, Lee & Vaithyanathan, 2002*). Table 8 shows the sample text of review before and after stemming.

The last step in the preprocessing phase is the removal of stop words. Stop words have no importance concerning the training of the classifiers. Instead, they increase the feature vector size and reduce the performance. So they must be removed to decrease the complexity of feature space and boost the training of classifiers. Table 9 shows the text of the sample review after the stopwords have been removed.

After the preprocessing is complete, feature extraction takes place where BoW, TF-IDF, and GloVe are used. Feature space for the sample reviews is given in Tables 10 and 11 for BoW and TF-IDF features, respectively. Experiments are performed with the standard dataset, as well as, the TextBlob annotated dataset to analyze the performance of the machine learning and proposed models.

The data are split into training and testing sets in a 75 to 25 ratio. Machine learning classifiers are trained on the training set while the test set is used to evaluate the

**Table 11 TF-IDF features from the preprocessed text of sample reviews.**

| No. | absolute | back | choice | come | didnt | do | great | gwyneth | like |
|---|---|---|---|---|---|---|---|---|---|
| 1 | 0.467351 | 0.000000 | 0.000000 | 0.000000 | 0.000000 | 0.000000 | 0.467351 | 0.467351 | 0.000000 |
| 2 | 0.000000 | 0.000000 | 0.346821 | 0.000000 | 0.346821 | 0.346821 | 0.000000 | 0.000000 | 0.346821 |
| 3 | 0.000000 | 0.408248 | 0.000000 | 0.408248 | 0.000000 | 0.000000 | 0.000000 | 0.000000 | 0.000000 |
| No. | movie | number | own | paltrow | show | tel | wish | would | |
| 1 | 0.355432 | 0.000000 | 0.000000 | 0.467351 | 0.000000 | 0.000000 | 0.000000 | 0.000000 | |
| 2 | 0.527533 | 0.346821 | 0.346821 | 0.000000 | 0.000000 | 0.000000 | 0.000000 | 0.000000 | |
| 3 | 0.000000 | 0.000000 | 0.000000 | 0.000000 | 0.408248 | 0.408248 | 0.408248 | 0.408248 | |

**Figure 3 Confusion matrix.**

performance of the trained models. For evaluating the performance, standard well-known parameters are used such as accuracy, precision, recall, and F1 score.

## Evaluation parameters

Performance evaluation of the classifiers requires evaluation metrics for which accuracy, precision, recall, and F1 score are selected concerning their wide use. The introduction of the confusion matrix is necessary to define the mathematical formulas for these evaluation metrics. The confusion matrix as shown in Fig. 3 can be considered as an error matrix that indicates four quantities. The confusion matrix shows true positive (TP), false positive (FP), true negative (TN), and false-negative (FN). Each row of the matrix represents the actual labels while each column represents predicted labels (*Landy & Szalay, 1993*).

TP indicates that the classifier predicted the review as positive and the original label is also positive. A review is TN if it belongs to the negative class and the real outcome is also negative. In the FP case, the review is predicted as positive, but the original label is negative. Similarly, a review is called FN if it belongs to the positive class but the classifier predicted it as negative (*Rokach & Maimon, 2005*).

Accuracy is a widely used evaluation metrics and indicates the ratio of true predictions to the total predictions. It has a maximum value of 1 for 100% correct prediction and the lowest value of 0 for 0% prediction. Accuracy can be defined as

**Table 12  Accuracy of the selected models with BoW features.**

| Classifier | Accuracy |
|---|---|
| DT | 0.72 |
| RF | 0.86 |
| GBC | 0.85 |
| SVM | 0.87 |

$$Accuracy = \frac{TP + TN}{TP + TN + FP + FN} \tag{7}$$

Precision focuses on the accuracy of predicting the positive cases. It shows what proportion of the positively predicted cases is originally positive. It is defined as

$$Precision = \frac{TP}{TP + FP} \tag{8}$$

Recall calculates the ratio of correct positive cases to the total positive cases. To get the ratio, the total number of TP is divided by the sum of TP and FN as follows

$$Recall = \frac{TP}{TP + FN} \tag{9}$$

F1 score is considered an important parameter to evaluate the performance of a classifier and has been regarded as more important than precision and recall. It defines how precise and robust is the classifier by incorporating precision and recall (*Bruce, Koger & Li, 2002*). F1 score value varies between 0 and 1 where 1 shows the perfect performance of the classifier. F1 score is defined as

$$F1\ Score = 2 \times \frac{precision \times recall}{precision + recall} \tag{10}$$

## RESULTS AND DISCUSSION

This study uses four machine learning classifiers to classify movie reviews into positive and negative reviews, such as DT. SVM, RF, and GBC. Four feature extraction approaches are utilized including TF-IDF, BoW, Word2Vec, and GloVe on the selected dataset to extract the features. Results for these feature extraction approaches are discussed separately. Similarly, the influence of TextBlob annotated data on the classification accuracy is analyzed. The contradictions in the sentiments expressed in the reviewers and the assigned sentiments cannot be ignored, so TextBlob is used to annotate the labels. Several experiments are performed using the standard, as well as, the TextBlob annotated dataset.

### Results using BoW features

Table 12 shows the classification accuracy of the machine learning classifiers when BoW features to train and test the classifiers. Results indicate that SVM can achieve an accuracy

**Table 13 Performance evaluation metrics using BoW features.**

| Model | Precision | | | Recall | | | F1 score | | |
|-------|------|------|--------|------|------|--------|------|------|--------|
|       | Pos. | Neg. | W avg. | Pos. | Neg. | W avg. | Pos. | Neg. | W avg. |
| DT  | 0.71 | 0.72 | 0.72 | 0.72 | 0.71 | 0.72 | 0.72 | 0.72 | 0.72 |
| RF  | 0.85 | 0.87 | 0.86 | 0.88 | 0.84 | 0.86 | 0.86 | 0.86 | 0.86 |
| GBC | 0.83 | 0.87 | 0.85 | 0.88 | 0.82 | 0.85 | 0.86 | 0.85 | 0.85 |
| SVM | 0.86 | 0.88 | 0.87 | 0.88 | 0.86 | 0.87 | 0.87 | 0.87 | 0.87 |

**Table 14 Accuracy of models with TF-IDF features.**

| Classifier | Accuracy |
|------------|----------|
| DT  | 0.71 |
| RF  | 0.86 |
| GBC | 0.86 |
| SVM | 0.89 |

of 0.87 with BoW features. Overall, the performance of all the classifiers is good except for DT whose accuracy is 0.72.

Performance of the classifiers is given in Table 13 in terms of precision, recall, and F1 score. The F1 score indicates that its value is the same with both positive and negative classes for all the classifiers, except for GBC who has F1 scores of 0.86 and 0.85 for positive and negative classes, respectively. Precision values are slightly different for positive and negative classes; for example, SVM has a precision of 0.88 and 0.90 for positive and negative classes. Similarly, although precision, recall, and F1 score of DT are the lowest but the values for positive and negative classes are almost the same. An equal number of the training samples in the dataset makes a good fit for the classifiers, and their accuracy and F1 scores are in agreement.

## Results using TF-IDF features

Table 14 contains the accuracy results for the classifiers using the TF-IDF features. It shows that the performance of the SVM has been elevated with an accuracy of 0.89 which is 2.29% higher than that of using BoW features. Unlike BoW which counts only the frequency of terms, TF-IDF also records the importance of terms by assigning higher weights to rare terms. So, the performance is improved when TF-IDF features are used as compared to BoW features.

Results for precision, recall, and F1 score are given in Table 15. Experimental results indicate that the F1 score is the same for positive and negative classes for all classifiers which indicates the good fit of the modes on the training data. On the other hand, precision for positive and negative classes is slightly different. For example, GBS has a precision of 0.84 and 0.87 while SVM has a precision of 0.88 and 0.90 for positive and negative classes, respectively.

**Table 15 Performance evaluation metrics using TF-IDF features.**

| Model | Precision | | | Recall | | | F1 score | | |
|---|---|---|---|---|---|---|---|---|---|
| | Pos. | Neg. | W avg. | Pos. | Neg. | W avg. | Pos. | Neg. | W avg. |
| DT | 0.72 | 0.71 | 0.71 | 0.70 | 0.72 | 0.71 | 0.71 | 0.71 | 0.71 |
| RF | 0.86 | 0.86 | 0.86 | 0.86 | 0.85 | 0.86 | 0.86 | 0.86 | 0.86 |
| GBC | 0.84 | 0.87 | 0.86 | 0.88 | 0.83 | 0.86 | 0.86 | 0.85 | 0.86 |
| SVM | 0.88 | 0.90 | 0.89 | 0.90 | 0.88 | 0.89 | 0.89 | 0.89 | 0.89 |

**Table 16 Performance of classifiers using GloVe features.**

| Model | Accuracy | Precision | | | Recall | | | F1 Score | | |
|---|---|---|---|---|---|---|---|---|---|---|
| | | Pos. | Neg. | W avg. | Pos. | Neg. | W avg. | Pos. | Neg. | W avg. |
| DT | 0.65 | 0.64 | 0.65 | 0.65 | 0.64 | 0.65 | 0.65 | 0.65 | 0.65 | 0.65 |
| RF | 0.74 | 0.75 | 0.74 | 0.74 | 0.72 | 0.77 | 0.74 | 0.73 | 0.75 | 0.74 |
| GBC | 0.65 | 0.65 | 0.65 | 0.65 | 0.65 | 0.66 | 0.65 | 0.65 | 0.65 | 0.65 |
| SVM | 0.75 | 0.75 | 0.75 | 0.75 | 0.75 | 0.75 | 0.75 | 0.75 | 0.75 | 0.75 |

SVM performs better for text classification than other supervised learning models, especially in the case of large datasets as this algorithm is derived from the theory of structural risk minimization (*Mouthami, Devi & Bhaskaran, 2013*).

## Classifiers results using GloVe features

Experimental results using GloVe features are shown in Table 16 for the selected classifiers. Results suggest that the performance of all the classifiers has been degraded when trained and tested on GloVe features. Glove features are based on the global word-to-word co-occurrence and count the co-occurred terms from the entire *corpus*. GloVe model is traditionally used with deep learning models where it helps to better recognize the relationships between the given samples of the dataset. In machine learning models, its performance is poor than that of TF-IDF features (*Dessi et al., 2020*). SVM and RF outperform other models using GloVE features.

## Results using Word2Vec features

Performance elevation metrics for all the classifiers using the Word2Vec features are given in Table 17. Results indicate that the performance of the classifiers is somehow better when trained and tested on Word2Vec features in comparison with GloVe features results. The performance of classifiers is not significant using Word2Vec features in comparison to the results of the classifiers using BoW and TF-IDF features. SVM achieved the highest accuracy of 0.88 with Word2Vec features as compared to other models because Word2Vec gives linear features set which is more suitable for SVM as compared to RF, GBC, and DT.

The comparison between machine learning models results on original dataset sentiment with BoW, TF-IDF, GloVe, and Word2Vec features are shown in Fig. 4. SVM is significant

**Table 17 Performance evaluation of classifiers using Word2Vec features.**

| Model | Accuracy | Precision | | | Recall | | | F1 Score | | |
|---|---|---|---|---|---|---|---|---|---|---|
| | | Pos. | Neg. | W avg. | Pos. | Neg. | W avg. | Pos. | Neg. | W avg. |
| DT | 0.65 | 0.65 | 0.65 | 0.65 | 0.65 | 0.65 | 0.65 | 0.65 | 0.65 | 0.65 |
| RF | 0.80 | 0.80 | 0.80 | 0.80 | 0.80 | 0.80 | 0.80 | 0.80 | 0.80 | 0.80 |
| GBC | 0.65 | 0.65 | 0.65 | 0.65 | 0.65 | 0.65 | 0.65 | 0.65 | 0.65 | 0.65 |
| SVM | 0.88 | 0.88 | 0.88 | 0.88 | 0.88 | 0.88 | 0.88 | 0.88 | 0.88 | 0.88 |

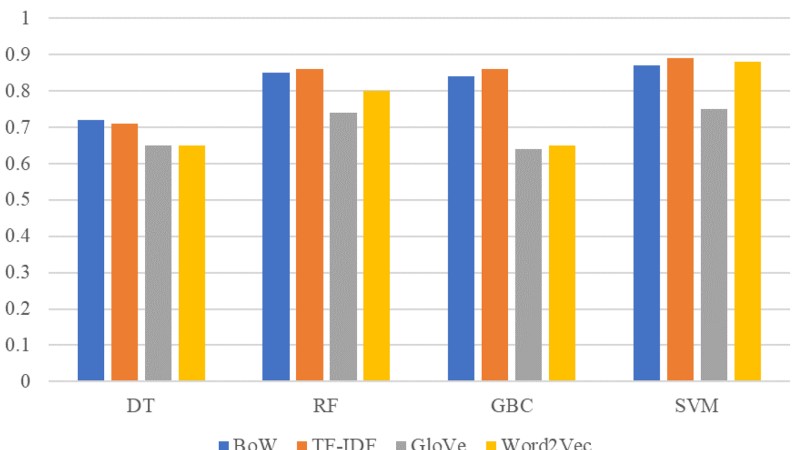

**Figure 4 Performance comparison between machine learning models using original dataset and BoW,TF-IDF, GloVe, Word2Vec features.**

with all features and achieved the best score with BoW, TF-IDF, and Word2Vec. This significant performance of SVM is because of its linear architecture and binary classification problem. SVM is more significant on linear data for binary classification with its linear kernel as shown in this study.

## Results on TextBlob annotated dataset

The contradictions in the users' expressed sentiments in the reviews and assigned labels can influence the sentiment classification accuracy of the models. To resolve this issue, TextBlob annotated data are used for the performance evaluation of the models. Results suggest that the performance of machine learning models is better when used with TextBlob labels than the original labels.

### Results using BoW features

The performance of models with BoW and TextBlob sentiments are shown in Table 18. Results indicate that SVM achieved its highest accuracy of the study 0.92 with TextBlob sentiments and BoW features. While the performance of other models such as RF, GBC, and DT has also been improved.

**Table 18 Performance evaluation of classifiers using BoW features on the TextBlob annotated dataset.**

| Model | Accuracy | Precision | | | Recall | | | F1 Score | | |
|---|---|---|---|---|---|---|---|---|---|---|
| | | Pos. | Neg. | W avg. | Pos. | Neg. | W avg. | Pos. | Neg. | W avg. |
| DT | 0.79 | 0.85 | 0.61 | 0.73 | 0.87 | 0.57 | 0.72 | 0.87 | 0.59 | 0.73 |
| RF | 0.85 | 0.84 | 0.90 | 0.87 | 0.98 | 0.47 | 0.72 | 0.90 | 0.62 | 0.76 |
| GBC | 0.82 | 0.85 | 0.70 | 0.78 | 0.92 | 0.55 | 0.73 | 0.98 | 0.62 | 0.75 |
| SVM | 0.92 | 0.94 | 0.84 | 0.89 | 0.94 | 0.84 | 0.89 | 0.94 | 0.84 | 0.89 |

**Table 19 Performance evaluation of classifiers using TF-IDF features on the TextBlob annotated dataset.**

| Model | Accuracy | Precision | | | Recall | | | F1 Score | | |
|---|---|---|---|---|---|---|---|---|---|---|
| | | Pos. | Neg. | W avg. | Pos. | Neg. | W avg. | Pos. | Neg. | W avg. |
| DT | 0.79 | 0.85 | 0.62 | 0.73 | 0.87 | 0.58 | 0.72 | 0.86 | 0.60 | 0.73 |
| RF | 0.84 | 0.85 | 0.88 | 0.87 | 0.98 | 0.51 | 0.74 | 0.91 | 0.65 | 0.78 |
| GBC | 0.83 | 0.86 | 0.73 | 0.79 | 0.92 | 0.57 | 0.75 | 0.89 | 0.64 | 0.77 |
| SVM | 0.92 | 0.92 | 0.88 | 0.90 | 0.96 | 0.78 | 0.87 | 0.94 | 0.82 | 0.88 |

**Table 20 Performance evaluation of classifiers using GloVe features on the TextBlob annotated dataset.**

| Model | Accuracy | Precision | | | Recall | | | F1 Score | | |
|---|---|---|---|---|---|---|---|---|---|---|
| | | Pos. | Neg. | W avg. | Pos. | Neg. | W avg. | Pos. | Neg. | W avg. |
| DT | 0.72 | 0.81 | 0.47 | 0.64 | 0.81 | 0.48 | 0.64 | 0.81 | 0.48 | 0.64 |
| RF | 0.80 | 0.71 | 0.81 | 0.76 | 0.94 | 0.39 | 0.67 | 0.87 | 0.51 | 0.69 |
| GBC | 0.72 | 0.81 | 0.47 | 0.64 | 0.81 | 0.48 | 0.65 | 0.81 | 0.48 | 0.64 |
| SVM | 0.81 | 0.83 | 0.71 | 0.77 | 0.93 | 0.46 | 0.70 | 0.88 | 0.56 | 0.72 |

*Results using TF-IDF features*

The performance of models with TF-IDF features and TextBlob sentiments are shown in Table 19. SVM achieves its highest accuracy score of 0.92 with TextBlob sentiment and TF-IDF features. While other models such as RF, GBC, and DT repeat their performances with TF-IDF features.

*Results using GloVe features*

Table 20 shows the performance comparison of models using GloVe features and TextBlob sentiments and it indicates that using GloVe features and TextBlob sentiments is better as compared to their performance on the original sentiments and GloVe features. Compared to the performance on the original dataset, the accuracy of the models has been improved significantly when used with TextBlob assigned sentiments. For example, the highest accuracy with GloVe features and TextBlob sentiment is 0.81 which was only 0.75

**Table 21 Performance evaluation of classifiers using Word2Vec features on the TextBlob annotated dataset.**

| Model | Accuracy | Precision | | | Recall | | | F1 Score | | |
|-------|----------|------|------|--------|------|------|--------|------|------|--------|
| | | Pos. | Neg. | W avg. | Pos. | Neg. | W avg. | Pos. | Neg. | W avg. |
| DT | 0.69 | 0.78 | 0.87 | 0.83 | 0.99 | 0.24 | 0.62 | 0.87 | 0.48 | 0.62 |
| RF | 0.79 | 0.78 | 0.87 | 0.83 | 0.99 | 0.24 | 0.62 | 0.87 | 0.38 | 0.63 |
| GBC | 0.70 | 0.80 | 0.44 | 0.62 | 0.80 | 0.44 | 0.62 | 0.80 | 0.44 | 0.62 |
| SVM | 0.88 | 0.90 | 0.83 | 0.87 | 0.95 | 0.71 | 0.83 | 0.92 | 0.77 | 0.84 |

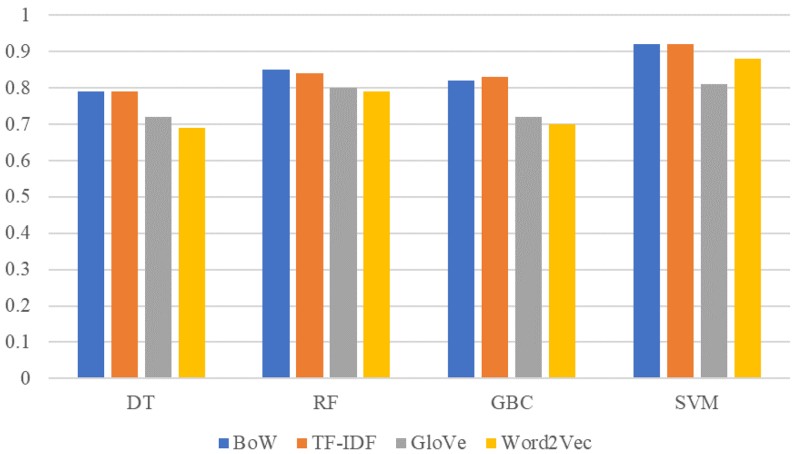

**Figure 5 Performance comparison between machine learning models using the TextBlob dataset and BoW,TF-IDF, GloVe, Word2Vec features.**

on the original sentiments. However, the performance of the machine learning models is inferior to that of BoW and TF-IDF.

### Results using Word2Vec features

Table 21 shows the performance of machine learning models with Word2Vec features and TextBlob sentiments. SVM achieves significantly better accuracy with Word2Vec features as compared to GloVe features. It gives a 0.88 accuracy score which is more than GloVe features but lower than BoW and TF-IDF features. RF and GBC achieve 0.79 and 0.70 accuracy scores, respectively. The performance of DT is degraded when used with Word2Vec features.

The comparison between machine learning model results on TextBlob sentiment dataset with BoW, TF-IDF, GloVe, and Word2Vec features is given in Fig. 5. SVM obtains better results with TextBlob sentiments using BoW and TF-IDF features as compared to GloVe and Word2Vec features. Similarly, the performance of other models such as RF, GBC, and DT has also been improved with the TextBlob sentiments. For the given dataset, the performance of models is good using TextBlob but these results can not be generalized for every other dataset. It is possible that a few original labels are not correct and using the TextBlob label can show better performance.

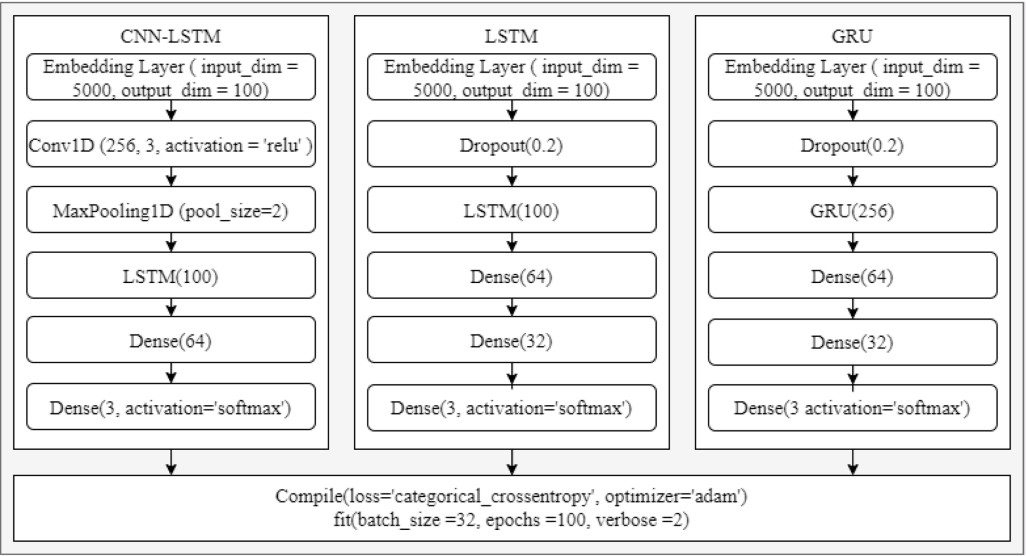

**Figure 6 LSTM, CNN-LSTM, and GRU architectures.**

The performance of machine learning models is good when used with TF-IDF features extracted from the original dataset and SVM outperforms with a significant 0.89 accuracy score. TF-IDF generates a weighted feature set as compared to BoW, GloVe, and Word2Vec features which helps to improve the accuracy of learning models. On the other hand, the accuracy of DT is reduced by 1% from 72% to 71% because DT is a rule-based model that performs well on simple term frequency as compared to weighted features. Weighted features introduce complexity in the DT learning process. SVM performs well because TF-IDF provides a linear feature set with the binary class which is more suitable for SVM that performs better being the linear model. The performance of machine learning models is improved with TextBlob data annotation. Machine learning models perform well with TF-IDF and BoW features and SVM obtains the highest accuracy of 0.92 accuracy score using TextBlob labels.

## Performance of deep learning models

To compare the performance of the proposed approach with the latest deep learning approach, experiments have been performed using several deep learning models. For this purpose, state-of-the-art deep learning models are used such as LSTM, CNN-LSTM, and Gated Recurrent Unit (GRU). The architecture of used deep learning models is provided in Fig. 6. These models are used with the TextBlob annotated dataset owing to the superior results on the dataset from machine learning models.

The performance of deep learning models is also good similar to machine learning models, as shown in Table 22. CNN-LSTM achieves a 0.90 accuracy score while GRU has a 0.86 accuracy. The results of deep learning models are shown in Table 6. Overall, the performance of deep learning models is slightly lower than the machine learning models. Regarding the machine learning model, SVM gives the highest accuracy of 0.92 while the

**Table 22 Performance analysis of deep learning models.**

| Model | Accuracy | Class | Precision | Recall | F1 Score |
|---|---|---|---|---|---|
| LSTM | 0.80 | Neg. | 0.83 | 0.79 | 0.81 |
| | | Pos. | 0.93 | 0.93 | 0.94 |
| | | Avg. | 0.88 | 0.87 | 0.87 |
| CNN-LSTM | 0.90 | Neg. | 0.78 | 0.88 | 0.83 |
| | | Pos. | 0.96 | 0.91 | 0.93 |
| | | Avg. | 0.87 | 0.90 | 0.88 |
| GRU | 0.86 | Neg. | 0.84 | 0.88 | 0.86 |
| | | Pos. | 0.88 | 0.83 | 0.85 |
| | | Avg. | 0.86 | 0.86 | 0.86 |

**Table 23 Performance analysis of the proposed methodology.**

| Year | Reference | Model | Accuracy |
|---|---|---|---|
| 2016 | *Sahu & Ahuja (2016)* | RF | 0.90 |
| 2017 | *Yenter & Verma (2017)* | CNN + LSTM | 0.895 |
| 2017 | *Giatsoglou et al. (2017)* | BoW-DOUBLE and Average emotion-DOUBLE | 0.83 |
| 2018 | *Mathapati et al. (2018)* | CNN | 0.89 |
| 2019 | *Ali, Abd El Hamid & Youssif (2019)* | CNN + LSTM | 0.89 |
| 2019 | *Bodapati, Veeranjaneyulu & Shaik (2019)* | LSTM + DNN | 0.885 |
| 2020 | *Tripathi et al. (2020)* | TF-IDF + LR | 0.891 |
| 2020 | *Qaisar (2020)* | LSTM | 0.899 |
| 2020 | *Shaukat et al. (2020)* | NN | 0.91 |
| 2021 | *Jain & Jain (2021a)* | CNN | 0.883 |
| 2021 | *Nafis & Awang (2021)* | SVM + (SVM-RFE) | 0.895 |
| 2021 | *Jain & Jain (2021b)* | NB + ARM | 0.784 |
| 2021 | Proposed | SVM + TextBlob + BoW & TF-IDF | 0.92 |

deep learning model CNN-LSTM achieves a 0.90 accuracy. The significant performance of machine learning models is because of handcrafted TF-IDF weighted features.

## Performance analysis with state-of-the-art approaches

Performance analysis has been carried out to analyze the performance of the proposed approach with other state-of-the-art approaches that utilize the IMDb movie reviews analysis. Comparison results are provided in Table 23. Results indicate that the proposed methodology can achieve competitive results to that of state-of-the-art approaches. The use of SVM with TF-IDF and BoW using the TextBlob technique provides an accuracy of 92% which is better than the state-of-the-art approaches.

## Statistical T-test

The *T*-test is performed in this study to show the statistical significance of the proposed approach (*Fatima et al., 2021*). The *T*-test is applied to SVM results with the proposed

**Table 24 Statistical *T*-test output values.**

| Student *T*-test output parameters | Output value |
| --- | --- |
| T-statistic | −0.182 |
| Critical value | 0.000 |

approach and original dataset. The output from the *T*-test favors either null hypothesis or alternative hypothesis.

- Accept Null Hypothesis: This means that the compared results are statistically equal.
- Reject Null Hypothesis: This means that the compared results are not statistically equal.

The output values of *T*-test in terms of T-statistic and critical Value are shown in Table 24. *T*-test infers that the null hypothesis can be rejected in favor of the alternative hypothesis because the T-statistic value is less than the critical value indicating that the compared values are statistically different from each other.

## CONCLUSION

With an ever-growing production of cinema movies, web series, and television dramas, a large number of reviews can be found on social platforms and movies websites like IMDb. Sentiment analysis of such reviews can provide insights about the movies, their team, and cast to millions of viewers. This study proposes a methodology to perform sentiment analysis on the movie reviews using supervised machine learning classifiers to assist the people in selecting the movies based on the popularity and interest of the reviews. Four machine learning algorithms including DT, RF, GBC, and SVM are utilized for sentiment analysis that is trained on the dataset preprocessed through a series of steps. Moreover, four feature extraction approaches including BoW, TF-IDF, GloVe, and Word2Vec are investigated for their efficacy in extracting the meaningful and effective features from the reviews. Results indicate that SVM achieves the highest accuracy among all the classifiers with an accuracy of 89.55% when trained and tested using TF-IDF features. The performance using BoW features is also good with an accuracy of 87.25%. Contrary to BoW which counts the occurrence of unique tokens, TF-IDF also records the importance of rare terms by assigning a higher weight to rare terms and perform better than BoW. However, the performance of the classifiers is greatly affected by GloVe and Word2Vec features which suggest that word embedding does not work well with the movie review dataset. For improving the performance of models and reducing the influence of contradictions found in the expressed sentiments and assigned labels, TextBlob is used for data annotation. Experimental results on TextBlob annotated dataset indicates that SVM achieves the highest accuracy of 92% with TF-IDF features. Compared to the standard dataset, the TextBlob assigned labels result in better performance from the models. The performance of deep learning models is slightly lower than machine learning models with the highest accuracy of 0.90 by the CNN-LSTM. Despite the equal number of positive and negative reviews used for training, the prediction accuracy for the positive and negative classes is different. Precision, recall, and F1 score indicate the models have a good fit, and

performance evaluation metrics are in agreement. The current study excludes the neutral class due to a low number of samples and experiments are performed using positive and negative classes. Consequently, the accuracy may have been higher as compared to that with three classes. Similarly, probable class imbalance by adding neutral class samples is not investigated and is left for the future. We intend to perform further experiments using movie reviews datasets from other sources in the future. Furthermore, the study on finding the contradictions in the sentiments expressed in the reviews and the assigned labels is also under consideration.

### Funding

This work was supported in part by the Basic Science Research Program through the National Research Foundation of Korea (NRF) funded by the Ministry of Education (NRF-2019R1A2C1006159 and NRF-2021R1A6A1A03039493), and in part by the 2021 Yeungnam University Research Grant. The funders had no role in study design, data collection and analysis, decision to publish, or preparation of the manuscript.

### Grant Disclosures

The following grant information was disclosed by the authors:
Ministry of Education: NRF-2019R1A2C1006159 and NRF-2021R1A6A1A03039493.
2021 Yeungnam University.

### Competing Interests

Imran Ashraf is an Academic Editor for PeerJ Computer Science.

### Author Contributions

- Muhammad Zaid Naeem conceived and designed the experiments, prepared figures and/or tables, and approved the final draft.
- Furqan Rustam conceived and designed the experiments, performed the computation work, authored or reviewed drafts of the paper, and approved the final draft.
- Arif Mehmood analyzed the data, authored or reviewed drafts of the paper, supervision, and approved the final draft.
- Mui-zzud-din performed the experiments, authored or reviewed drafts of the paper, and approved the final draft.
- Imran Ashraf analyzed the data, authored or reviewed drafts of the paper, and approved the final draft.
- Gyu Sang Choi analyzed the data, authored or reviewed drafts of the paper, funding, and approved the final draft.

### Data Availability

The dataset is available at Kaggle: https://www.kaggle.com/lakshmi25npathi/imdb-dataset-of-50k-movie-reviews.

The code is available in the Supplemental Files.

## Supplemental Information

Supplemental information for this article can be found online at http://dx.doi.org/10.7717/peerj-cs.914#supplemental-information.

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
