# Peer review of "Classification of movie reviews using term frequency-inverse document frequency and optimized machine learning algorithms"

_PeerJ Computer Science, doi:10.7717/peerj-cs.914_

## Round 0.1 · original submission · Major Revisions

The paper sounds interesting, but all the reviewers agreed it needs some improvements. Please prepare a new manuscript accordingly.

Reviewer 1 ·

Basic reporting

no comment

Experimental design

no comment

Validity of the findings

no comment

Additional comments

Given the complexity involved, the author has produced a number of positive and welcome outcomes including the literature review which offers a useful overview of current research and policy and the resulting bibliography which provides a very useful resource for current practitioners.

Most relevant papers (only relevant and state-of-the-art) should be qualitatively and quantitatively analyzed with what gaps were left in these works and what this work is proposed to overcome those gaps/challenges.

Experimental should be rigorously analyzed both theoretically and visually with proper justification of obtained results as well as with potential comparative studies

can add future work also

Reviewer 2 ·

Basic reporting

This study provides the implementation of various machine learning models to measure
the polarity of the sentiment presented in user reviews on IMDB website. For this purpose, the reviews are first preprocessed to remove redundant information and noise and then various classification models like support vector machines (SVM), Naive Bayes classifier, random forest and gradient boosting classifier are used to predict the sentiment of these reviews. The objective is to find the optimal process and approach to attain the highest accuracy with best generalization. Various feature engineering approaches such as term frequency-inverse document frequency (TF-IDF), bag of words, global vectors for word
representations and Word2Vec are applied along with the hyperparameter tuning of the classification models to enhance the classification accuracy

Experimental design

Satisfactory

Validity of the findings

Satisfactory

Additional comments

1. The authors have to summarize the findings/gaps from recent literature in the form of a table.
2. Some of the recent works on NLP and feature extraction such as the following can be discussed: "An ensemble machine learning approach through effective feature extraction to classify fake news, Analysis of dimensionality reduction techniques on big data".
3. In the proposed methodology section, the authors should clearly map the proposed work with the limitations of the existing works and discuss how the proposed work overcomes them.
4. Present a detailed analysis on the results obtained.
5. Discuss about the limitations of the current work.

Reviewer 3 ·

Basic reporting

Literature review is very superficial and the last paper that is reviewed has been published in the year 2019.
Literature review is not guided, and authors have made very generic assumptions.

Experimental design

Authors are advised to try the latest LSTM models that are more suitable for this type of problem.

Validity of the findings

There isn’t any significant contribution in terms of methodology or results.
Authors need to justify the results, why a particular model should yield better results.

Additional comments

The paper is basically a comparative analysis of various machine learning algorithms on the published dataset and has no real scientific contribution.

Annotated reviews are not available for download in order to protect the identity of reviewers who chose to remain anonymous.

Reviewer 4 ·

Basic reporting

No comment.

Experimental design

1.There is no novelty in this research. The algorithms used (for preprocessing and classification) are common
algorithms that widely used for sentiment analysis.
2. The third contribution mentions the use of deep learning in this work which is not correct since the results only show four traditional ML algorithms.
3. Authors should specify the type of decision tree used in this paper and also other parameters such as maximum depth of the tree or number of features to consider when looking for the best split.
4. Authors should also specify the hyperparameters used and the kernels used in this paper.

Validity of the findings

1. The results in table 15 show comparable results in previous work. Hence, the impact and novelty of this work are not there. The author should indicate the strength of this work.
2. The use of "optimized machine learning algorithms" term in the title does not reflect the real research outcome. The authors should at least describe the optimization process involved in this work.
3. In terms of the evaluation validity, how the authors ensure the similar evaluation method were used by all the papers in Table 15?

---

## Round 0.2 · Major Revisions

One of the reviewers has serious objections to this version, so I recommend you to prepare a new one addressing all the changes suggested.

Reviewer 2 ·

Basic reporting

The authors have addressed all my comments. The manuscript may be accepted for publication.

Experimental design

Satisfactory

Validity of the findings

No comments

Additional comments

The authors have addressed all my comments. The manuscript may be accepted for publication.

Reviewer 5 ·

Basic reporting

The paper presents a study to find the optimal process and approach to measure the polarity of the sentiments in user reviews on the IMDB website.

In section "Related Work" authors present a wide research in the context of the gap of knowledge that they want to fill. However, two of the works in Table 1 are not reviewed in the text above: Giatgoglou et al (2017) and Mathapati et al (2018).

I thank you for providing the code, however your code file need more comments to be useful to future readers. For example, the introduction doesn't match with the code (Multi-layer Perceptron is not used, according to the paper). Besides, there aren't any comments about the kind of features engineering method used in each point.

The way of citing the bibliographic references is not correct the most of the times. The cites must be between parentheses if order to make the text more readable. For example, in line 48, instead of "...as a significant research are during the last few years Hearst (2003)" should be "...as a significant research are during the last few years (Hearst, 2003)" The kind of cite, using the name of the author outside of the parentheses, should only be used if the name of author is part of the sentence. For example: Singh et al (2013) conduct experimental work about...

Some minor errors are:
a) Bag of Words is abbreviate as BoW, but sometimes it is written as Bow. Authors should revise the text to homogenize this acronym and others (RF)
b) Some blank spaces are needed in lines 196 (between algorithms and Oghina), 201 (between consistency and Lee)
c) Equations must be referenced with their number between parentheses instead of saying "the following". For example, in lines 219 and 220, TF-IDF is calculated using TF and IDF as (3)
d) The sample reviews of the Table 5, 6, 7 and 8 contain point at the end of the sentences, when these were removed in Table 4
e) In Table 6, the first sample review after case lowering yet contains a "P"
f) In line 325, it is indicated that Tables 11 and 13 show BoW and TF-IDF features, but the tables are 9 and 10.

Experimental design

In line 181, bibliography reference to the data set is missing

A bibliography reference to TextBlow is missing. It is a Python Library for processing textual data that provides a simple API for diving into common natural language processing (NLP) tasks such as sentiment analysis between others. It is not a lexicon-based technique.

TextBlob is used to annotate the dataset with sentiments (negative or positive) in order to tackle with "the contradiction in the user sentiments in the reviews and assigned labels" How do the authors know that classification of the reviews is contradictory and TextBlob result is better? Authors must explain this point.

In line 197, what does authenticity of a learning algorithm mean? or it is a mistake and it should be "accuracy".

In subsection "Feature Engineering Methods" only Bag of Words and TF-IDF are explained. Authors should also explain and reference the other two methods used in their study (Global Vector and Word2Vec)

Authors should revise the equation (1) and its explanation in line 217. In it TFij appears, but the explanation talk about term "t" and document "d"

In subsection "Supervised machine learning Models", author explain Naive Bayes, but in the Introduction and in the results, they used Gradient Boosting Classifier. Since gradient boosting classifiers are a group of machine learning algorithms that combine many weak learning models together to create a strong predictive model, which weak learning models does the gradient boosting classified used combine?

The title of the paper contains "optimized machine learning algorithms" and Table 3 shows the hyperparameters used for optimizing the performance of models. Author should explain how and why they have chosen these parameters to ensure the use of optimized machine learning algorithms.

The BoW features showed in Table 9 don't match with those in the preprocessed text of sample reviews in Table 8

TF-IDF values in Table 10 are miscalculated. If a word have a a frequency of 0 in a document, its TF-IDF can't be upper 0

In Figure 3, rows of the matrix represent actual labels and columns represent predicted labels. However, in lines 337 and 338 the opposite appears

Validity of the findings

The results are not very conclusive and the analysis of them it is merely a reading of the table with no discussion or a deep explanation.

There are not statics test to see if the differences are significant

---

## Round 0.3 · Major Revisions

The reviewer has brought up the same issues once again. Please take these suggestions seriously and address them in the next version of the paper.

Reviewer 5 ·

Basic reporting

The authors have addressed all my comments.

Experimental design

Comment 10: A bibliography reference to TextBlow is missing. It is a Python Library for processing textual data that provides a simple API for diving into common natural language processing (NLP) tasks such as sentiment analysis between others. It is not a lexicon-based technique.

The bibliography reference to TextBlow must indicate that it is a online document, the url and the last query date.


Author Response comment 11: Reviewer’s is concern is genuine. For clarification we added Table 3 in the revised manuscript
which contains the ’original’ label from the dataset and label from the Textblob. Original label is ’Negative’ while Textblob assigned label is ’positive’. Textblob assigned labels are more reliable as is confirmed by the higher classification accuracy from the models when used with Textblob labeled dataset.

It make no sense to rely on the classification results to indicate that the labels in the original dataset are contradictory. I'm sorry but this point is not clear for me. Why do we have to give more confidence to a machine than to a human? I'll try to set this clear. From what I read, I understand that you are using the predictions from another tool (Textblob) to train and evaluate your proposal. In this way, you can say that your proposal is similar to a model created by Textblob but it may be a poor model to the real data. In fact, in this way Textblob model could not be improved because that model is the goal. So if that model is supposed to be perfect, what's the point of the research (we already have the model from Textblob). So the main question is: Why do you say the labels from Textblob are better than the original data? I am sorry to say that Table 3 is not clarifying because the text seems to have been processed and can not be interpreted. How do we know if it is positive or negative?

Validity of the findings

In the new discussion of results, the paper says:
"Performance of machine learning models(*1*) is improved with Textblob data annotation(*2*)"

In the response to comment 11:
*2* "Textblob assigned labels(*2*) are more reliable as is confirmed by the higher classification accuracy(*1*)"

This is a case of false proof. If we want to prove *1*, we need to prove *2*. If we want to prove *2*, we need to prove *1*. We enter in a recursive non-sense proof.

Regarding this, the conclusions talk about contradictions found between sentiments and assigned labels but this is not explained in the paper. Is the data set wrongly labeled? You seem to mean that Textblob is detecting this mislabeled instances but how has this been checked? Has anybody actually read the reviews and confirmed that they was mislabeled? If not, you can not say so because it will just create confusion, in a similar way as fake news do.

Besides, more care should be taken when writing. If the data set is about movie reviews, why mentioning "tweets" in the discussion.

The statistical test applied is not clearly explained. What differences has been tested? T-test is usually applied to see if the means are different. What two means have been compared?

---

## Round 0.4 · accepted · Accept

After checking the reviewers' opinions, I consider the paper ready for publication.

Reviewer 5 ·

Basic reporting

The authors have addressed my comments.

Experimental design

The authors have addressed my comments.

Validity of the findings

The authors have addressed all my comments.